# Systematic Review and Meta-Analysis on Prevalence and Antimicrobial Resistance Patterns of Important Foodborne Pathogens Isolated from Retail Chicken Meat and Associated Environments in India

**DOI:** 10.3390/foods14040555

**Published:** 2025-02-07

**Authors:** Haris Ayoub, Murthy Suman Kumar, Zunjar Baburao Dubal, Kiran Narayan Bhilegaonkar, Hung Nguyen-Viet, Delia Grace, Sakshi Thapliyal, Ekkoruparambil Sethurajan Sanjumon, Elisetty Naga Pavana Sneha, Dharavath Premkumar, Vinodh Kumar Obli Rajendran, Ram Pratim Deka

**Affiliations:** 1International Livestock Research Institute, National Agricultural Science Complex, Pusa, New Delhi 110012, India; harisayoub018@gmail.com (H.A.); sakshi61214@gmail.com (S.T.); 2Division of Veterinary Public Health, ICAR-Indian Veterinary Research Institute, Izatnagar 243122, India; sumanvph@gmail.com (M.S.K.); drzunjar@yahoo.co.in (Z.B.D.); kiranvph@rediffmail.com (K.N.B.); sanju.sethuraj231@gmail.com (E.S.S.); nagapavanasneha@gmail.com (E.N.P.S.); 3International Livestock Research Institute, P.O. Box 30709, Nairobi 00100, Kenya; h.nguyen@cgiar.org (H.N.-V.); d.randolph@cgiar.org (D.G.); 4Natural Resources Institute, University of Greenwich, Central Avenue, Chatham ME4 4TB, UK; 5Division of Epidemiology, ICAR-Indian Veterinary Research Institute, Izatnagar 243122, India; drdpremkumar402@gmail.com

**Keywords:** food safety, public health, retail chicken meat

## Abstract

The chicken value chain, a vital part of the global food supply, also represents a significant public health concern due to the risk of foodborne pathogens, particularly in low- and middle-income countries (LMICs) such as India. This systematic review and meta-analysis aimed to assess the prevalence of significant bacterial pathogens including *Salmonella* spp., *Campylobacter* spp., *Escherichia coli*, *Staphylococcus aureus*, *Listeria monocytogenes*, *Clostridium perfringens*, and *Klebsiella pneumonia*. in retail chicken meat and associated environments and the antimicrobial resistance based on the articles published between January 2010–December 2023. The research adhered to the guidelines in the ’Preferred Reporting Items for Systematic Review and Meta-Analysis’ (PRISMA). Based on 90 included studies, *S. aureus* showed the highest pooled prevalence (56%; 95% CI: 38–74%), followed by *E. coli* (50%; 95% CI: 37–64%), *C. perfringens* (35%; 95% CI: 10–65%), and *K. pneumoniae* (21%; 95% CI: 7–38%). *Salmonella* spp. (95% CI: 11–26%) and *Campylobacter* spp. (95% CI: 11–27%) exhibited similar prevalence rates at 18%, while *L. monocytogenes* had the lowest prevalence at 13% (95% CI: 1–33%). A sensitivity analysis was subsequently conducted to assess the impact of influential studies, and the pooled prevalence of each pathogen was recalculated after removing these studies to ensure the robustness of the results. The pathogens, specifically *Salmonella* spp. and *Campylobacter* spp., displayed high levels of resistance to medically important antimicrobials (erythromycin, tetracycline, ciprofloxacin, colistin), a potential threat to human health. This study advocates for a collaborative and comprehensive approach, reflecting the multifaceted nature of the issue, and highlighting the importance of a holistic strategy to safeguard public health and maintain antibiotic effectiveness in the face of emerging challenges.

## 1. Introduction

In India, chicken is the most preferred meat as it is affordable, widely available, and culturally acceptable [1]. The demand for chicken meat is rising, driven by higher incomes, urbanization, and the vertical integration of the poultry industry [2]. With a poultry population of 851.81 million, India is the fifth largest global poultry producer and the sixth largest in poultry meat production, reflecting the sector’s rapid expansion [3,4]. Approximately 3.77 million tons of poultry meat are consumed annually in India, with an annual per capita consumption of 2.38 kg [5]. Chicken meat is nutrient-dense and recommended in the Indian dietary guidelines [6]. However, the intensification of poultry farming and widespread antimicrobial use in the sector have raised concerns about the prevalence of foodborne pathogens and the potential for antimicrobial resistance development in bacteria associated with chicken meat [7].

Foodborne disease (FBD) is a major public health issue with a global health burden comparable to that of malaria, tuberculosis, or HIV/AIDS; nearly two-thirds of this burden is due to bacterial pathogens [8]. Most of this burden is borne by low- and middle-income countries (LMICs), with children less than five years disproportionately affected. In India, FBD is responsible for nearly 100 million illnesses a year and 117,000 deaths [9]. Most foodborne diseases are attributable to animal-source food and fresh produce [10,11]. Foodborne pathogens represent a significant concern, especially in poultry products such as chicken meat. Pathogens such as *Salmonella* spp., *Campylobacter* spp., and *Clostridium perfringens* are commonly found in these products and have been associated with numerous foodborne illnesses worldwide [12]. The emergence and spread of these pathogens in poultry complicate efforts to manage infections, as they often exhibit resistance to multiple antibiotics, posing challenges not only to human health but also to veterinary practices [13].

*Salmonella* spp. are a leading cause of foodborne gastroenteritis, characterized by symptoms such as diarrhea, abdominal cramps, and fever [14]. The contamination of poultry carcasses with *Salmonella* is widely documented, with non-host-specific strains capable of causing food poisoning in humans [15]. In the U.S., antibiotic-resistant *Salmonella* strains are associated with tens of thousands of illnesses annually, highlighting the need for effective intervention strategies in the poultry supply chain [16].

*Campylobacter* spp. are another critical group of pathogens commonly found in poultry. They are a leading cause of bacterial gastroenteritis globally and are primarily associated with the consumption of undercooked or improperly handled poultry [17]. Even at low levels, *Campylobacter* can cause infections, and the pathogen’s growing resistance to antibiotics such as fluoroquinolones and macrolides has further complicated treatment options [18,19]. This resistance, particularly in *Campylobacter jejuni* and *Campylobacter coli*, poses significant challenges for both clinical management and food safety interventions.

*Escherichia coli* is another major foodborne pathogen found in poultry meat, acting as a critical indicator of overall food safety and hygiene practices. Certain pathogenic strains, such as Shiga toxin-producing *E. coli* (STEC), are associated with severe illnesses, including hemorrhagic colitis and hemolytic uremic syndrome [20]. The rising prevalence of antibiotic-resistant *E. coli* strains in poultry not only threatens food safety but also represents a potential reservoir of resistance genes that could be transmitted to humans [21]. The presence of plasmid-mediated colistin resistance in *E. coli*, crucial for treating carbapenem-resistant infections, necessitates robust surveillance and control measures [22].

Pathogens such as *Staphylococcus aureus* (including MRSA), *Listeria monocytogenes*, and *Clostridium perfringens* are also frequently associated with poultry meat. *Staphylococcus aureus* is a common cause of staphylococcal food intoxication, characterized by a rapid onset of symptoms following the ingestion of contaminated food [23]. MRSA poses significant challenges due to its resistance to multiple classes of antibiotics, including β-lactams and fluoroquinolones, complicating treatment and increasing the risk of zoonotic transmission, particularly in individuals with close animal contact [24]. *Listeria monocytogenes*, notable for its resilience to refrigeration and high fatality rates, particularly among immunocompromised individuals, presents another critical concern due to its increasing resistance to commonly used antibiotics [25].

The complexity of managing foodborne pathogens in poultry is further compounded by the presence of emerging pathogens like *Clostridium perfringens* and *Klebsiella pneumoniae*, which are increasingly reported in poultry products. *Clostridium perfringens* is among the top pathogens causing foodborne illnesses in the U.S., with resistance patterns that vary significantly by region [26]. *Klebsiella pneumoniae*, an opportunistic pathogen, is prevalent in various environments and exhibits diverse resistance mechanisms, making it a significant concern in both clinical and food safety contexts [27]. The emergence of multidrug-resistant *Klebsiella pneumoniae* (MDR-KP) and carbapenem-resistant *Klebsiella pneumoniae* (CRKP) has been highlighted as a critical public health issue, with infection-related fatality rates ranging from 40% to 70% [28,29].

Antimicrobial resistance (AMR) is a formidable global health challenge [30]. In G20 nations, including the Russian Federation, China, and India, over 40% of infections are attributed to resistant bacteria, much higher than the 17% in Organisation for Economic Co-operation and Development (OECD) countries [31]. In India, AMR was directly attributed to 297,000 deaths and associated with another 1,042,500 deaths in 2019 alone [32]. In 2020, antimicrobial use in food-producing animals was estimated at 99,502 tons, with India among the top five consumers [33]. The extensive use of antibiotics in agriculture, particularly in livestock, contributes to this huge health burden through multiple routes, including contact with animals, contact with environments contaminated with animal waste, consumption of crops contaminated with animal waste, and consumption of animal products [34].

Despite the global and national significance of this issue, there is a lack of comprehensive data on the prevalence and AMR patterns of foodborne pathogens in the poultry value chain in India. While several systematic reviews have been conducted globally, highlighting the prevalence of resistant pathogens in poultry, there remains a critical gap in the Indian context, where agricultural practices and antibiotic use differ significantly from other regions. This systematic review and meta-analysis investigate the prevalence of significant bacterial pathogens isolated from chicken meat in India and the patterns of antimicrobial resistance they exhibit. By synthesizing data from diverse studies conducted across the country, the focus is on providing a nuanced understanding of the current state of antimicrobial resistance in these key pathogens. This knowledge should be crucial for evidence-based policymaking, guiding antimicrobial stewardship efforts, and ultimately safeguarding public health and the sustainability of India’s poultry industry.

## 2. Materials and Methods

### 2.1. Study Design and Search Strategy

The search strategy included the period from January 2010 to December 2023 using electronic databases, viz., Google Scholar, Science Direct, Springer, PubMed, ResearchGate, Krishikosh, and ICAR-CeRA. The search strategy utilized Boolean operators and keywords including ‘chicken meat’, ‘foodborne pathogens’, ‘prevalence’, ‘occurrence’ ‘antimicrobial resistance’, and ‘India’. After the results were viewed for a particular pathogen in the ‘foodborne pathogen’ search, the same was included as a keyword, viz., ‘*Salmonella*’, ‘*Campylobacter*’, ‘*Listeria monocytogenes*’, ‘*Staphylococcus aureus*’, ‘*Escherichia coli*’, ‘*Clostridium perfringens*’ and ‘*Klebsiella pneumoniae*’. An iterative approach was implemented to refine the search strategy based on initial results, ensuring a comprehensive and systematic identification of relevant literature for the systematic review and meta-analysis.

### 2.2. Inclusion, Exclusion Criteria and Quality Check of the Reviewed Literatures

Criteria for the inclusion and exclusion of studies are outlined in Table 1. The quality criteria for inclusion used checklists from the Meta-analysis of Observation Studies in Epidemiology (MOOSE), 2000 [35], and the Preferred Reporting Items for Systematic Reviews and Meta-Analyses Protocols, 2015 [36]. The results of the review were reported using the Preferred Reporting Items for Systematic Reviews and Meta-Analyses (PRISMA) 2020 checklist [37].

Two independent investigators, namely HA and ST, conducted a manual screening of studies identified by the search. Where discrepancies arose between the two investigators, resolution was by a third investigator, VOR. The reliability and validity of risk of bias assessment scores were evaluated using the Newcastle–Ottawa Scale (NOS) as endorsed by the Cochrane Collaboration [38]. The process for study inclusion is shown in Figure 1.

### 2.3. Data Extraction

Full texts were obtained for data extraction, and names, publication year, study location, total samples, number of samples from different sources, number of isolates for each pathogen, and antimicrobial susceptibility data were organized in Microsoft Excel^®^. The extracted data were entered in a separate excel sheet for each pathogen. For the prevalence estimation, systematic review and meta-analysis was carried out. However, after data extraction, it was observed that the data regarding the AMR of the pathogens were not sufficient to conduct the meta-analysis; hence, for AMR, only the systematic review was carried out.

### 2.4. Meta-Analysis

Publication bias was assessed by visualizing the symmetry of the funnel plot, rank correlation, and Egger’s test. Cochran’s Q and Higgins’ I^2^ [39] methods were used to evaluate the heterogeneity within the study. Values of I^2^ exceeding 75% were regarded as indicating a high level of heterogeneity [40]. The meta-analysis utilized the inverse-variance model [41] and the Freeman–Tukey double arcsine transformation [42,43]. The pooled estimate for each pathogen was reported as prevalence for that pathogen with a 95% confidence interval (CI) and prediction interval (PI). Forest plots were employed to visually depict the prevalence in each study and the aggregated estimated prevalence. Visual examination of Baujat plots [44] was employed to assess between-study heterogeneity and identify outlier studies.

To identify influential studies, diverse case deletion diagnostics such as covariance ratio (COVRATIO), studentized residuals, Cook’s distances, the difference in fit values (DFFITS), and leave-one-out estimates were utilized [45].

A leave-out-one sensitivity analysis was conducted to assess the impact of each study on the pooled prevalence of each pathogen in chicken meat and its associated environment, while gradually excluding each study. Statistical analyses were performed using the R statistical platform (R Foundation for Statistical Computing, Vienna, Austria, version 3.5.1) with the “meta” and “metafor” packages.

## 3. Results

### 3.1. Search Results and Study Characteristics

The search process yielded a total of 149 full-text articles related to pathogens in the poultry value chain. On screening the articles at title and abstract level, two duplicates for *Campylobacter* spp. were excluded. On full-text screening, 57 articles were further excluded based on reasons like no clear methodology and insufficient data, as well as not describing prevalence or AMR in any of the bacteria. The included studies covered all regions of India and all nodes of the chicken value chain.

### 3.2. Descriptive Analysis of All Included Studies

The articles included were categorized into retail chicken meat, chicken meat products, and chicken-associated environments (encompassing slaughter and production environments, as well as cloacal/faecal and intestinal samples). The analysis included 90 studies focusing on the prevalence of various pathogens within the chicken value chain. The details of the published articles included in the systematic review and meta-analysis is given in Table 2.

### 3.3. Prevalence of Pathogens in Poultry Meat

The prevalence of significant foodborne pathogens in India was estimated using a total of 90 studies; 29 studies for *Salmonella* spp. [46,47,48,49,50,51,52,53,54,55,56,57,58,59,60,61,62,63,64,65,66,67,68,69,70,71,72,73,74], 29 for *Campylobacter* spp. [51,59,75,76,77,78,79,80,81,82,83,84,85,86,87,88,89,90,91,92,93,94,95,96,97,98,99], 20 for *E. coli* [56,63,64,66,68,74,91,100,101,102,103,104,105,106,107,108,109,110], 11 for *S. aureus* [51,59,62,64,65,66,73,74,111,112,113], nine for *L. monocytogenes* [56,59,73,95,114,115,116,117,118], six for *C. perfringens* [59,65,119,120,121,122], and five for *K. pneumonia* [74,107,108,123,124]. The overall prevalence of significant bacterial pathogens in retail chicken meat and associated environment is depicted in forest plots.

Funnel plot, Regression Test, and Rank Correlation Test identified the publication bias in most of the pathogens (Table 3). Significant heterogeneity was evident among the study observations, based on Cochrane Q and I^2^ statistics. Baujat plots were used to the detect studies which contribute to the heterogeneity in the meta-analysis (Appendix A). The outlier and influential study analysis results are shown in Table 3.

Results from leave-one-out sensitivity analysis showed that the combined effects did not significantly change as a result of the excluded study. The prevalence of significant and emerging bacterial pathogens in retail chicken meat and associated environment from India is given in Table 3.

### 3.4. Pooled Prevalence of Salmonella spp.

The pooled prevalence of *Salmonella* spp. from the retail chicken meat and associated environment was 18% (95% CI: 11–26%). The corresponding forest plot is given in Figure 2. There was a high heterogeneity between studies (I^2^ = 98%, *p* < 0.01) (Table 3). The study Kumar et al., 2020 [59] was identified as an outlier as well as influential study. Excluding Kumar et al., 2020 [59], the random effects model (REM) pooled prevalence estimate (PPE) for *Salmonella* spp. was 15.47% (95% CI 9.73; 22.27%). For retail chicken meat, the pooled prevalence for *Salmonella* spp. was 20% (95% CI: 12–30%), and the study by Kumar et al., 2020 [59] was an outlier and influential study, excluding which the PPE was 17.27% (95% CI: 10.64–25.04%). For chicken-associated environments, the pooled prevalence was 13% (95% CI: 4–27%). In this analysis, Ramya et al., 2012 [71] was identified as an outlier and influential study. Omitting Ramya et al., 2012 [71], the PPE was 7.68% (95% CI: 3.16–13.65%).

### 3.5. Pooled Prevalence of Campylobacter spp.

The pooled prevalence estimates and heterogeneity of studies for *Campylobacter* spp. from the retail chicken meat and associated environments (encompassing all studies), retail chicken meat, and associated environments are shown in Table 3 and depicted in forest plots in Figure 3. Including all studies, Begum et al., 2015 [90], and Bobade et al., 2022 [98], were outlier studies, and Bobade et al., 2022 [98], was an influential study, omitting which the PPE was 15.77% (95% CI: 9.83–22.76%). For chicken meat, Khan et al., 2018 [99], was an outlier and influential study, omitting which the PPE was 16% (95% CI: 7–28%). In relation to chicken-associated environments, Begum et al., 2015 [90]; Tayde et al., 2014 [84]; and Rajendran et al., 2012 [83], were identified outliers, and Tayde et al., 2014 [84], were influential, omitting which the PPE was 17.14% (95% CI: 9.47–26.46%).

### 3.6. Pooled Prevalence of E. coli

The aggregated prevalence estimates and heterogeneity of studies pertaining to *E. coli* in the retail chicken meat and associated environments (encompassing all studies), retail chicken meat, chicken meat products, and the associated environments are presented in Table 3 and Figure 4. Among all the studies, Kumar et al., 2014 [63]; Deshmukh et al., 2023 [110]; Vaidya et al., 2016 [65]; Shaikh, 2015 [91]; and Saikia and Joshi, 2010 [56], were identified as outlier studies. None of the studies were influential. Within the studies examining the prevalence of *E. coli* in retail chicken meat, Kumar et al., 2014 [63]; Deshmukh et al., 2023 [110]; Hussain et al., 2017 [101]; Shaikh, 2015 [91]; Kaushik et al., 2018 [68]; Kumar et al., 2020 [59]; Kumar et al., 2021 [102]; and Saikia and Joshi, 2010 [56], were classified as outlier studies. Notably, none of these studies were influential. In the context of the chicken-associated environment, Deshmukh et al., 2023 [110], was identified as outlier and influential study, omitting which the PPE was 40% (95% CI: 34.33–45.80%). Regarding studies on meat products, Giri et al., 2021 [107]; Tamta, 2022 [108]; and Anukampa et al., 2020 [109], were all outlier studies, and Tamta, 2022 [108], and Anukampa et al., 2020 [109], were influential studies. Excluding the two studies, the PPE was 27% (95% CI: 6–61%).

The systematic review showed the prevalence of *E. coli* with an array of virulent genes in chicken meat and its associated environment. For instance, 10 out of 62 *E. coli* isolates carried the *stx2* virulence gene, identifying them as Shiga toxin-producing *E. coli* (STEC), a known pathogenic type, as reported by Kaushik et al., 2018 [68]. Similarly, study by Saikia and Joshi, 2014 [105] detected 22 isolates as STEC-positive, with 11 (50%) some carrying the *eae* factor, further indicating their pathogenic potential. In another study, Bhave et al. (2019) [100] identified 19 of the 146 isolates as extraintestinal pathogenic *E. coli* (ExPEC) through PCR screening. Other researchers have detected *E. coli* isolates lacking pathogenic markers, suggesting that although these isolates are not directly pathogenic to humans, they could serve as indicators of contamination.

### 3.7. Pooled Prevalence of C. perfringens

The combined prevalence estimates and heterogeneity across studies for *C. perfringens* in the retail chicken meat and associated environments and retail chicken meat are depicted in Table 3. The corresponding forest plot is presented in Figure 5. Considering all included studies, Kumar et al., 2020 [59], was identified as an outlier study and influential study. When excluding Kumar et al., 2020 [59], PPE was estimated as 24.87% (95% CI: 11.4–41.35%). In the context of studies focusing on chicken meat, Kumar et al., 2020 [59], was pinpointed as both an outlier and an influential study. Upon excluding Kumar et al., 2020 [59], the PPE was 24.69% (95% CI: 5.51–51.35%).

### 3.8. Pooled Prevalence of Listeria spp.

The forest plot in Figure 6 shows the pooled prevalence and heterogeneity of studies regarding *Listeria* spp. from chicken meat. The pooled prevalence was 13% (95% CI: 1–33%). Kumar et al., 2020 [59], was identified as outlier and influential study, excluding which the PPE was 7.03% (95% CI: 0.4–20.41%).

### 3.9. Pooled Prevalence of S. aureus and K. pneumoniae

The forest plot in Figure 7 depicts the combined prevalence and heterogeneity of studies concerning *S. aureus* in chicken meat. Meti et al., 2022 [51]; Kumar et al., 2020 [59]; Badhe et al., 2013 [62]; and Thanigaivel et al., 2015 [64], were recognized as outlier studies. However, none of the studies were influential.

The estimated pooled prevalence and heterogeneity across the studies focusing on *K. pneumoniae* within the retail chicken meat and associated environments and chicken meat are shown in Table 3. The corresponding forest plots are depicted in Figure 8. Considering all included studies for the prevalence of *K. pneumoniae*, Tewari et al., 2019 [123], was identified as both an outlier and an influential study. Upon excluding Tewari et al., 2019 [123], the pooled prevalence estimate (PPE) was 10.48% (95% CI: 7.77–13.49%). Pertaining to retail chicken meat, Anukampa et al., 2020 [109], and Tamta et al., 2022 [108], were outliers and influential studies. Excluding the two studies, the PPE of *K. pneumoniae* in chicken meat was 11.81% (95% CI: 7.32–17.11%).

### 3.10. AMR of Bacterial Pathogens from Retail Chicken Meat and Its Associated Environments

A systematic review of AMR in bacterial pathogens from retail chicken meat and its associated environments was conducted. However, a meta-analysis was not performed due to the variability in antibiotic susceptibility testing methodology and the insufficient number of studies reporting AMR for specific pathogen-antibiotic combinations.

### 3.11. Antimicrobial Resistance (AMR) of Gram-Negative Bacteria

A systematic review of AMR in Gram-negative bacterial pathogens isolated from retail chicken meat and its associated environments highlighted significant resistance trends across multiple studies. The findings revealed widespread multidrug resistance (MDR), with alarming resistance profiles for key pathogens:

*Salmonella* spp.: A total of 131 *Salmonella* spp. isolates from three studies demonstrated pervasive MDR. High resistance rates were observed across multiple antibiotics, with universal resistance to tetracycline and erythromycin. In North India, all 70 isolates were MDR, with over 92% showing resistance to five distinct classes of antibiotics [53]. Additionally, 31 isolates from retail chicken meat and broiler farms exhibited 100% resistance to at least three or more antimicrobial classes [54].

*Campylobacter* spp.: *Campylobacter* spp. isolates, totaling 439 from 10 studies, displayed widespread MDR, with 94% of 101 *C. jejuni* isolates exhibiting resistance to two or more antibiotics. Universal resistance to nalidixic acid was documented in all 14 *C. jejuni* isolates from chicken meat samples [99]. High rates of MDR were consistently reported, with resistance levels ranging from 54.37% to 97% in various studies.

*E. coli*: A total of 397 *E. coli* isolates from six studies revealed diverse resistance profiles. Universal resistance to specific antibiotics such as ampicillin, colistin, and nitrofurantoin was observed in certain studies. For example, 77 *E. coli* isolates from retail chicken meat exhibited universal resistance to these antibiotics [103]. Variable resistance was noted among 62 isolates, with the highest resistance documented against cefuroxime and penicillin [68].

Overall, Gram-negative bacteria exhibited significant MDR, particularly against β-lactams, fluoroquinolones, and tetracyclines, raising critical concerns for public health and food safety. The consistently high resistance levels to fluoroquinolones and β-lactams among Gram-negative bacteria align with global trends, as these classes are heavily used in human and veterinary medicine.

### 3.12. Antimicrobial Resistance (AMR) of Gram-Positive Bacteria

Gram-positive bacterial pathogens isolated from retail chicken meat and the associated environment displayed concerning levels of MDR, albeit less extensive compared to Gram-negative bacteria. Key findings include:

*S. aureus*: A total of 80 *S. aureus* isolates from one study demonstrated resistance to multiple antibiotics, including ampicillin, tetracycline, and erythromycin [111]. Resistance to ampicillin was universal, while moderate resistance was observed for tetracycline (up to 87.5%) and erythromycin.

*C. perfringens*: Overall, 63 *C. perfringens* isolates from one study exhibited high rates of MDR, with notable resistance to key antibiotics such as linezolid, clarithromycin, and erythromycin [122]. Despite the high resistance rates, *C. perfringens* isolates were fully susceptible to ofloxacin, offering a potential therapeutic option for infections caused by this pathogen.

In general, Gram-positive bacteria showed relatively lower MDR levels than Gram-negative bacteria but demonstrated resistance to critical antibiotics such as macrolides and β-lactams, posing significant challenges in clinical settings. The universal susceptibility of *S. aureus* to vancomycin and *C. perfringens* to ofloxacin are an option of treatment for these pathogens.

Important AMR findings for all study pathogens are summarized in Table 4. Resistance pattern for each pathogen is shown graphically in Figure 9.

## 4. Discussion

To the best of our knowledge, the present systematic review and meta-analysis is the most recent, comprehensive attempt to synthesize and analyse data regarding the prevalence and antibiotic resistance patterns of important foodborne pathogens within retail chicken meat and its associated environments in India, focusing on *Salmonella* spp., *Campylobacter* spp., *E. coli*, *S. aureus*, *L. monocytogenes*, *C. perfringens*, and *K. pneumoniae*.

The systematic review and meta-analysis of the collated data from the published studies (2011–2023) all across India depicted the prevalence of important foodborne pathogens in retail chicken meat and its surroundings. Adherence the study to the MOOSE and PRISMA guidelines ensured the higher quality of the outcome.

*E. coli* has often been considered a marker organism for faecal contamination and hygiene [125]. Non-pathogenic isolates of *E. coli* might reflect the hygiene status within the poultry chain and indicate faecal contamination or non-compliance in general with the principles of food safety [126]. Moreover, different pathotypes of *E. coli* create a health risk to consumers due to their virulent and multidrug-resistant traits, causing limited antimicrobial therapies to function effectively [127,128]. This systematic review highlights the widespread presence of *E. coli* with diverse virulence gene profiles in chicken meat and associated environments, presenting a significant public health concern. Certain virulent strains, such as Shiga toxin-producing *E. coli* (STEC), are known to cause severe illness in humans. For example, Kaushik et al., 2018 [68] reported that 10 out of 62 *E. coli* isolates carried the stx2 gene, a key marker of STEC pathogenicity. Similarly, Saikia and Joshi (2014) [105] identified 22 (11.5%) STEC-positive isolates, with some carrying the eae gene, which enhances their ability to adhere to host cells and increases their pathogenic potential. These findings highlight the risk of zoonotic transmission from poultry to humans, particularly in settings with poor hygiene practices.

The detection of extraintestinal pathogenic *E. coli* (ExPEC) by Bhave et al., (2019) [100] with 19 isolates identified through PCR screening, points to poultry as a reservoir for strains capable of causing serious infections in humans, such as urinary tract infections and neonatal meningitis. In contrast some researchers have reported *E. coli* isolates without pathogenic markers. Although these isolates may not pose a direct threat to human health, they serve as indicators of fecal contamination, which underscores lapses in hygiene or cross-contamination during meat processing.

These findings highlight the urgent need for robust biosecurity measures, such as routine surveillance for virulent *E. coli*, improved hygiene practices in poultry production and slaughter, and public health initiatives aimed at reducing contamination risks.

The prevalence varied among the study pathogens, with *S. aureus* having the highest pooled prevalence (56%), followed by *E. coli* (50%), *C. perfringens* (35%), and *K. pneumoniae* (21%). *Salmonella* spp. and *Campylobacter* spp. exhibited similar prevalence rates at 18%, while *L. monocytogenes* had the lowest prevalence at 13%. During the slaughter process, cross-contamination from the skin and mucous membranes of slaughtered animals, as well as lapses in hygiene control, can lead to microbial contamination. For example, studies by Althaus et al., 2017 [129] and Zweifel et al., 2014 [130] have highlighted the importance of a process analysis that identifies operations in the slaughtering process where contamination can increase or decrease. These studies emphasized that the application of risk-based preventive measures, such as the HACCP approach [131], is crucial for controlling contamination and ensuring food safety. Proper temperature control and prevention of cross-contamination are vital in mitigating the risks of pathogens like *S. aureus*, which thrive under inadequate refrigeration conditions. A somewhat similar trend was observed in a meta-analysis of published studies in Europe. The meta-analysis showed *S. aureus* (38.5%) exhibited the highest prevalence among the four pathogens, followed by *Campylobacter* spp. (33.3%). However, *Salmonella* spp. prevalence (7.10%) was lower than that of *L. monocytogenes* (19.3%) [13].

A high prevalence of *E. coli* (45.16%), similar to our study, has been reported in a meta-analysis of the prevalence of micro-organisms in chicken from the UK [132]. This high prevalence may be attributed to contamination of water used for different purposes in the chicken value chain that may contaminate the carcass. Another meta-analysis revealed a high prevalence of *E. coli* from water in Africa (overall prevalence: 71.7%; drinking water: 61.9%) [133].

A higher pooled prevalence of *Campylobacter* spp. from chicken was reported (31.88%) from a UK study [132]. However, the lower prevalence in our study may be due to variations in sampling methodologies and laboratory techniques employed between the studies.

The high heterogeneity observed in prevalence estimates (Table 3) can be attributed to several factors. Differences in geographical regions, such as urban versus rural settings, influence pathogen prevalence. Sample types also play a role, with variations between retail shops and slaughterhouses impacting contamination levels. Seasonal factors, particularly during warmer months, facilitate bacterial growth, leading to higher prevalence rates. Additionally, variability in testing protocols and laboratory methods contributes to discrepancies. While subgroup analyses could clarify these sources of heterogeneity, the lack of detailed covariate data across studies limited our ability to conduct such analyses. This highlights the need for standardized reporting in future prevalence studies. High levels of AMR in bacterial pathogens imply a significant public health risk, necessitating a concerted effort towards responsible antimicrobial use and surveillance in the food production chain. *Salmonella* spp. and *Campylobacter* spp. isolates consistently demonstrated high levels of multidrug resistance, posing a significant challenge to effective treatment. The resistance to critical antibiotics such as ciprofloxacin and cefotaxime in *Salmonella* spp. raises concerns about potential threats to human health. Similarly, *Campylobacter* spp. isolates show resistance to commonly used antibiotics like tetracycline and ciprofloxacin, with a notable prevalence of multidrug resistance. *E. coli* exhibits diverse resistance patterns, emphasizing the need for alternative therapeutic strategies. *S. aureus* isolates from chicken meat display high resistance rates to several antibiotics, highlighting the urgency for careful antibiotic management. *C. perfringens* isolates exhibit resistance against multiple antibiotics, and the prevalence of multidrug resistance calls for comprehensive monitoring and intervention strategies.

Resistance in *Salmonella* spp. to critical antibiotics such as ciprofloxacin and cefotaxime has been associated with the production of extended-spectrum beta-lactamases (ESBLs), which are increasingly prevalent in poultry isolates [134,135]. Horizontal gene transfer, particularly through plasmids carrying resistance genes such as *blaCTX-M* and *qnr*, plays a crucial role in the dissemination of resistance in poultry environments [136,137].

The high prevalence of bacterial pathogens in retail chicken meat, along with the significant antimicrobial resistance (AMR) in these bacteria, pose a major public health concern in India. This dual threat is amplified by the widespread misuse of antibiotics in poultry farming, which accelerates the development of resistant strains. Implementing control measures at every stage of the poultry production chain is crucial to mitigating these risks. Effective biosecurity practices at farms can significantly reduce the pathogen burden before chicken meat reaches consumers. Measures such as vaccination against common pathogens like *Salmonella* and the use of probiotics as alternatives to antibiotics can play a pivotal role in controlling infection at its source [138,139]. Implementing strict hygiene protocols, sanitation, and limiting access to farms can prevent the introduction and spread of harmful bacteria, while reducing the reliance on antibiotics [140]. Equally important is the need for robust surveillance systems that monitor antibiotic usage and resistance patterns. India requires a national-level surveillance program that tracks antibiotic use across poultry farms and enforces strict regulations to minimize overuse. Responsible antibiotic use should be promoted, accompanied by research into alternative interventions such as bacteriophage therapy and prebiotics, which have been shown to reduce dependency on conventional antibiotics [64,66]. Another critical area for intervention is improving cold chain infrastructure. Inadequate refrigeration during storage and transportation facilitates the growth of bacteria such as *L. monocytogenes*, contributing to foodborne illnesses [141]. This is particularly concerning in rural areas, where cold chain systems may be weak or absent [142]. Investing in cold chain logistics will help ensure poultry products remain safe from contamination until they reach consumers, significantly reducing the risk of infections. Lastly, raising public awareness is key to mitigating the spread of foodborne pathogens and AMR. Educational campaigns targeted at farmers, retailers, and consumers must emphasize the importance of food safety, proper hygiene, and cooking practices to reduce the incidence of infections. Public engagement on these issues could foster behavior change across the poultry supply chain, leading to better compliance with food safety regulations and reduced AMR.

The Food Safety and Standards Authority of India (FSSAI) guidelines mandate temperatures from 0 °C to 5 °C during transportation to inhibit bacterial growth [143]. Hygiene regulations stipulate sanitation standards for slaughterhouses and retail outlets to minimize contamination risks [144]. Despite these regulations, significant implementation gaps exist, particularly in informal markets where cold chain infrastructure is often lacking [145]. Strengthening enforcement mechanisms and investing in cold chain logistics could significantly mitigate foodborne pathogen risks.

Overall, addressing the AMR challenge requires a coordinated approach, incorporating farm-level interventions, regulatory policies, infrastructure improvements, and public education to safeguard human health.

## 5. Conclusions

The systematic review and meta-analysis reveal the prevalence of *S. aureus*, *E. coli*, *Campylobacter* spp., and *Salmonella* spp., in retail chicken meat and its associated environments. Alarmingly, these pathogens exhibit widespread resistance to critical antibiotics, including ciprofloxacin and tetracycline, with multidrug resistance being common among *Salmonella* spp. and *Campylobacter* spp. isolates. The findings signify the substantial public health risk posed by poultry products in the country and warrants the urgent need for a One Health approach and coordinated interventions, including antimicrobial stewardship, improved biosecurity measures, and robust surveillance systems. Effective strategies such as vaccination, antibiotic alternatives, and enhanced cold chain infrastructure are critical to mitigating contamination and reducing AMR. Furthermore, consumer education on proper food handling and cooking practices is essential to curbing the spread of foodborne pathogens.

## Figures and Tables

**Figure 1 foods-14-00555-f001:**
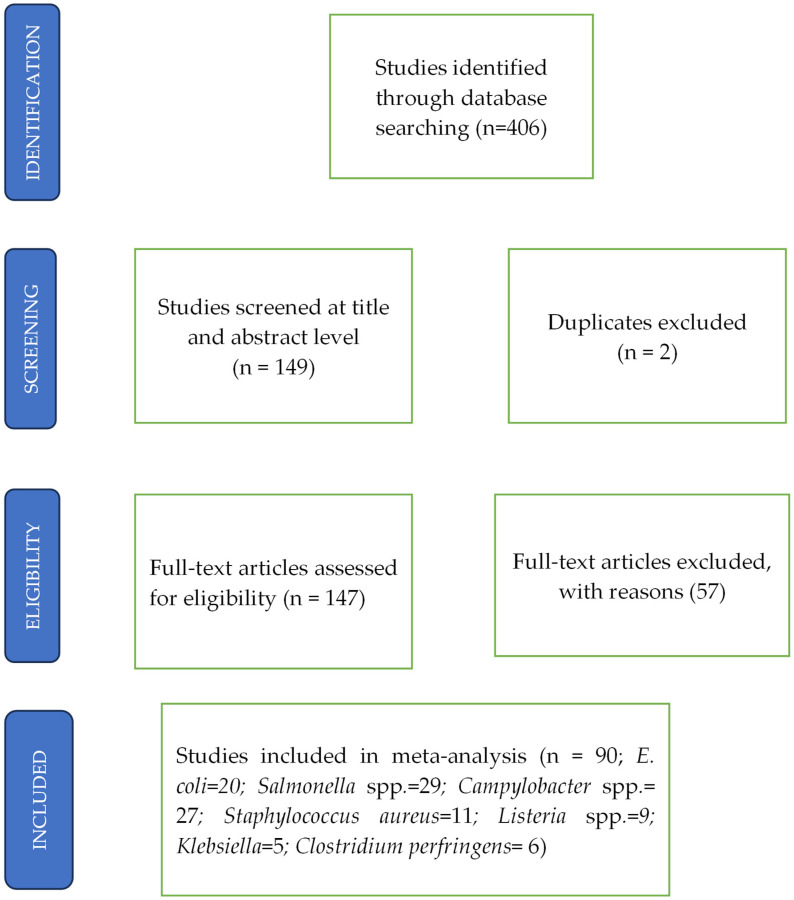
PRISMA schematic diagram depicting the method utilized to conduct this systematic review.

**Figure 2 foods-14-00555-f002:**
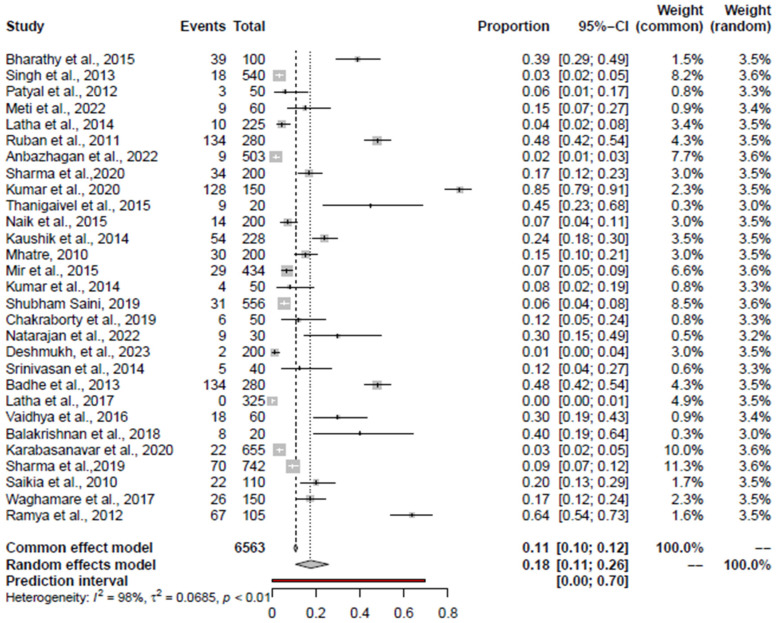
Forest plot depicting the prevalence of *Salmonella* spp. in the chicken value chain in India reported for each included publication in the meta-analysis [46,47,49,50,51,52,53,54,55,56,57,58,59,60,61,62,63,64,65,66,68,70,71,72,73,74,110,111]. Weightage given to each included publication by both RE and FE models have been shown for comparison. “Total” refers to the number of samples tested in each publication, “Events” refers to the number of positive samples, and “Proportion” refers to the prevalence for each publication. The prediction interval is marked in red line.

**Figure 3 foods-14-00555-f003:**
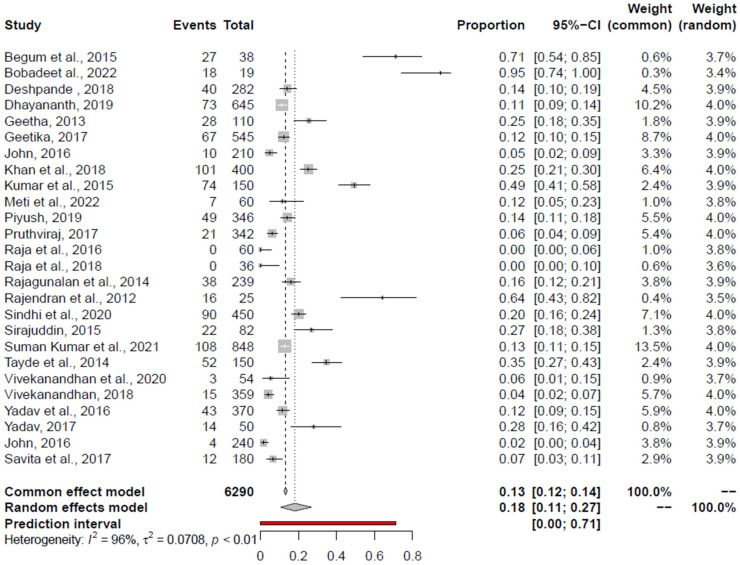
Forest plot depicting the pooled prevalence of *Campylobacter* spp. [18,51,59,75,77,78,79,80,81,82,84,85,86,87,88,89,90,91,92,94,95,96,97,98,99] in the chicken value chain.

**Figure 4 foods-14-00555-f004:**
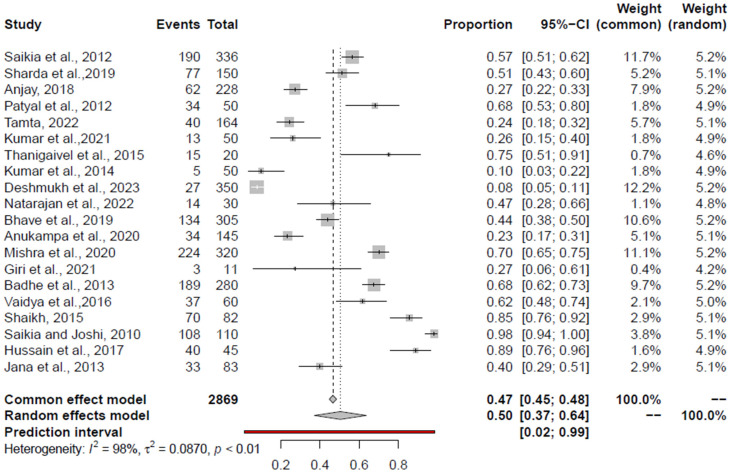
Forest plot depicting the pooled prevalence of *E. coli* in the chicken value chain [56,62,63,64,65,66,68,74,91,100,101,102,103,105,106,107,108,109,110].

**Figure 5 foods-14-00555-f005:**
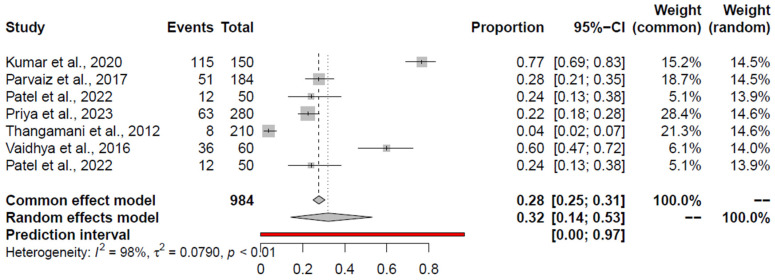
Forest plot depicting the pooled prevalence of *C. perfringens* in the chicken value chain [59,65,119,120,121,122].

**Figure 6 foods-14-00555-f006:**
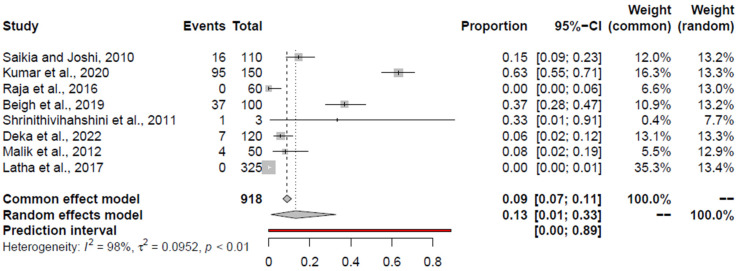
Forest plot depicting the pooled prevalence of *Listeria* spp. in the retail chicken meat [56,59,73,95,115,116,117,118].

**Figure 7 foods-14-00555-f007:**
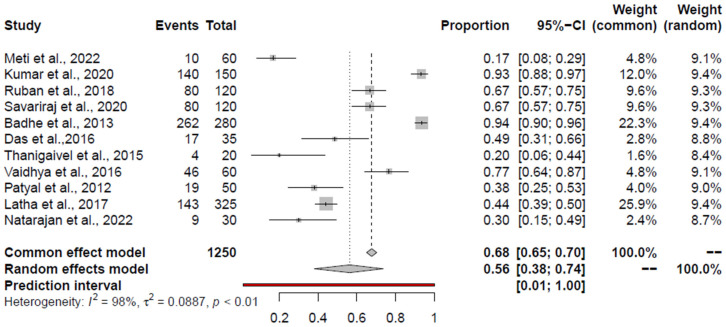
Forest plot depicting the pooled prevalence of *S. aureus* in the retail chicken meat [51,59,62,64,65,66,73,74,111,112,113].

**Figure 8 foods-14-00555-f008:**
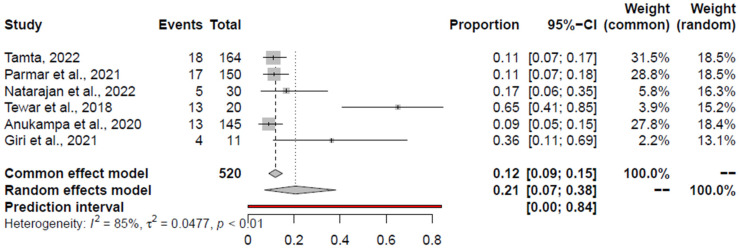
Forest plot depicting the pooled prevalence of *K. pneumoniae* in the chicken value chain [74,107,108,109,123,124].

**Figure 9 foods-14-00555-f009:**
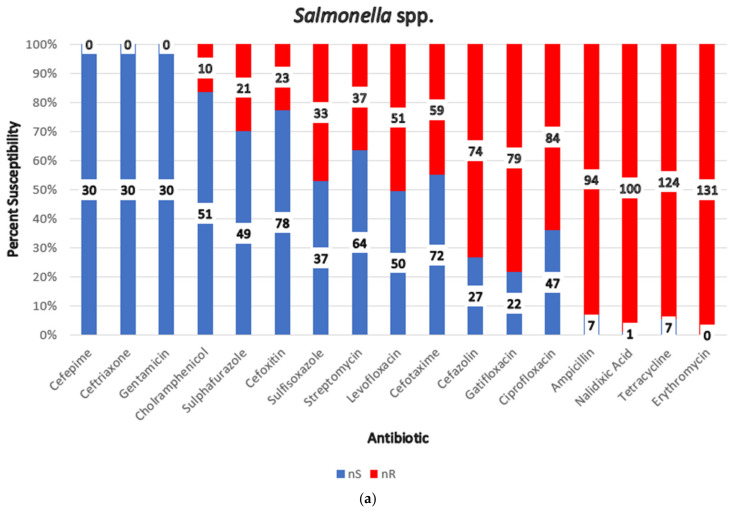
The antibiotic susceptibility pattern of different pathogens to the indicated antibiotics (nR = No. of resistant isolates, nS = No. of non-resistant isolates, nS + nR = Total no. of isolates tested for a particular antibiotic): (**a**) *Salmonella* spp., (**b**) *Campylobacter* spp., (**c**) *E. coli*, (**d**) *S. aureus*, (**e**) *C. perfringens*.

**Table 1 foods-14-00555-t001:** Inclusion and exclusion criteria used for the systematic review and meta-analysis of the prevalence of foodborne pathogens in chicken value chain (2010–2023) in India.

Inclusion Criteria	Exclusion Criteria
Studies on chicken meat associated pathogens between 2010 and 2023	Studies before 2010 or after 2023
Studies on foodborne pathogens in India	Studies on foodborne pathogens in countries other than India
Studies with clear methodologies and sampling procedures	Studies with insufficient or unclear methodology and incomplete outcome data
Conducted within the chicken food value chain, covering production, processing, distribution, and consumption stages	Research conducted outside the chicken food value chain
Studies reporting on occurrence, prevalence, and AMR in bacterial pathogens	Studies lacking data on occurrence, prevalence, and AMR
Full-text availability of chosen studies	Exclusive focus on non-bacterial pathogens or exploration of pathogens unrelated to chicken meat in the food value chain
Articles or theses available in English	Articles without full-text availability or not in English

**Table 2 foods-14-00555-t002:** No. of included studies in systematic review and meta-analysis of each pathogen for prevalence and AMR analysis.

Bacteria	Prevalence Studies	AMR Studies
*Salmonella* spp.	23 (retail chicken meat), 14 (chicken-associated environment)	3
*Campylobacter* spp.	16 (retail chicken meat), 17 (chicken-associated environment)	10
*E. coli*	21 (retail chicken meat), 4 (chicken-associated environment), 3 (meat products)	6
*S. aureus*	11 (retail chicken meat)	1
*L. monocytogenes*	9 (retail chicken meat)	-
*C. perfringens*	5 (retail chicken meat), 2 (chicken-associated samples)	1
*Klebsiella* spp.	5 (retail chicken meat), 3 (meat products), 2 (chicken-associated samples)	-

**Table 3 foods-14-00555-t003:** Meta-analysis of prevalence of pathogens in chicken value chain in India (RCM + AE = retail chicken meat and associated environment; encompasses all types of samples related to production and slaughter environment, retail chicken meat, and meat products and eggs; AE = associated environment; RCM = retail chicken meat; CMP = chicken meat products).

Pathogen	Sample Type/Category	Percent Pooled Prevalence (95% CI)	Influential Studies	Percent Pooled Prevalence After Removal of the Influential Study (95% CI)	Heterogeneity (I^2^)	Between-Study Variance (τ²)	Regression Test (*p* Value)	Rank Correlation Test (*p* Value)
*Salmonella* spp.	RCM + AE	18 (11; 26)	Kumar et al., 2020 [59]	15.47 (9.73; 22.27%)	0.98	0.0685	0.0046	0.0447
	RCM	20 (12; 30)	Kumar et al., 2020 [59]	17.27 (10.64; 25.04)	0.98	0.0677	-	0.1942
	AE	13 (04; 27)	Ramya et al., 2012 [71]	7.68 (3.16; 13.65)	0.96	0.0871	0.0116	0.015
*Campylobacter* spp.	RCM + AE	18 (11; 27)	Bobade et al., 2022 [98]	15.77 (9.83; 22.76)	0.96	0.0708	<0.0001	0.0001
	RCM	17 (8; 28)	Khan et al., 2018 [99]	16 (7; 28)	0.94	0.0714	<0.0001	0.0001
	AE	21 (11; 33)	Rajendran et al., 2012 [83]	17.14 (9.47; 26.46).	0.96	0.0689	<0.0001	0.0003
*E. coli*	RCM + AE	50 (37; 64)	None	-	0.98	0.0870	0.8496	0.9745
	RCM	57 (43; 71)	None	-	0.97	0.0949	0.1917	0.9235
	AE	28 (11; 49)	Deshmukh et al., 2023 [110]	40 (34.33; 45.80).	0.97	0.0425	0.3006	0.75
	CMP	7 (0; 27)	Giri et al., 2021 [107] Anukampa et al., 2020 [109]	-	0.83	0.0440	0.0243	1
*C. perfringens*	RCM + AE	35 (10; 65)	Kumar et al., 2020 [59]	24.87 (11.4; 41.35)	0.99	0.0790	-	0.3567
	RCM	32 (14; 53)	Kumar et al., 2020 [59]	24.69 (5.51; 51.35)	0.98	0.1154	0.4866	0.8167
*K. pneumoniae*	RCM + AE	21 (7; 38)	Tewari et al., 2019 [122]	10.48 (7.77; 13.49)	0.85	0.0477	<0.0001	0.0167
	RCM	13 (8; 19)	None	-	0.46	0.0030	0.1652	0.75
*Listeria* spp.	RCM	13 (1; 33)	Kumar et al., 2020 [59]	7.03 (0.4; 20.41)	0.98	0.0952	0.3851	0.1789
*S. aureus*	RCM	56 (38; 74)	None	-	0.98	0.0887	0.0047	-

**Table 4 foods-14-00555-t004:** Significant AMR findings for important foodborne pathogens (Gram-negative pathogens: *Salmonella* spp., *Campylobacter* spp., *E. coli*; Gram-positive pathogens: *S. aureus*, *C. perfringens*).

Bacteria	Study	No. of Isolates	Significant Findings (Resistance %)	Remarks
*Salmonella* spp.	Sharma et al., 2019 [53]	70	Nalidixic acid (98.57%). ampicillin (95.71%), ciprofloxacin (82.86%), gatifloxacin (81.43%)	Every isolate in the study was multidrug-resistant. Over 92% of isolates were resistant to five antibiotic classes. Tetracycline and erythromycin showed universal resistance (100%).
	Mhatre, 2010 [57]	30	100% sensitivity to cefotaxime, cefepime, ceftriaxone, chloramphenicol, ciprofloxacin, and gentamicin; 100% resistance to erythromycin and tetracycline.	
	Saini, 2019 [54]	31	Ampicillin (87.09%), ciprofloxacin (83.87%), tetracycline (77.42%), cefotaxime (74.19%), gatifloxacin (70.97%)	Erythromycin and nalidixic acid 100% resistance
*Campylobacter* spp.	Khan et al., 2018 [99]	101	Co-trimoxazole (84.1%), cephalothin (81.1%), tetracycline (59.4%)	97% overall resistance, 94% multidrug resistance
	Suman Kumar et al., 2021 [82]	103	Tetracycline (64.1%), doxycycline (54.4%), ampicillin (46.3%), nalidixic acid (42.7%)	54.37% multidrug resistance, common resistance in chicken meat
	Pruthviraj, 2017 [78]	23	Amikacin (26.08%), tetracycline (17.39%)	Majority sensitive to most drugs, 4.34% resistance to erythromycin
	Deshpande, 2018 [81]	31	Tetracycline (87.09%), ciprofloxacin (70.96%), nalidixic acid (38.70%)	
	Yadav, 2017 [77]	14	Nalidixic acid (100%), ampicillin (85.72%), ciprofloxacin (42.86%)	Varying resistance patterns observed
	Yadav et al., 2016 [94]	43	Polymyxin-B (100%), chloramphenicol (97.67%), gentamicin (95.35%)	Complete resistance to penicillin-G, methicillin, rifampicin
	Dhayananth, 2019 [75]	40	Cefoxitin (95%), ciprofloxacin (80%), nalidixic acid (25%)	Various resistance patterns observed
	Garhia, 2017 [87]	42	Cefoxitin (97.61%), ciprofloxacin (64.28%), nalidixic acid (33.33%)	Majority resistant to cefoxitin
	Vivekanandhan, 2018 [89]	13	Oxacillin, tetracycline, cefpodoxime (84.61% each), ciprofloxacin (69.23%)	Resistance levels varied among antibiotics
	Begum et al., 2015 [93]	27	Amoxicillin, co-trimoxazole (100%), cephalexin (96.29%)	Majority sensitive to gentamicin, intermediate ciprofloxacin
*E. coli*	Singh et al., 2019 [103]	77	Ampicillin, colistin, nitrofurantoin (100%), cefixime (80.52%), co-trimoxazole (72.7%)	Widespread drug resistance observed; 87% sensitivity to amikacin, 100% sensitivity to chloramphenicol
	Senapati et al., 2020 [104]	224	Oxytetracycline (64.73%), chloramphenicol (58.48%), ampicillin/cloxacillin (57.14%), ciprofloxacin (77.68%)	Diverse resistance patterns, significant susceptibility to cefepime and imipenem (about 94%)
	Kaushik et al., 2018 [68]	62	Cefuroxime, penicillin (89.1% each), ampicillin (80.43%), vancomycin (74.1%), ciprofloxacin (76%)	Diverse resistance patterns, 87% susceptibility to amikacin and gentamicin, 93% to ciprofloxacin
	Jana and Mondal, 2013 [106]	13	Novobiocin (100%), cefixime, sulphafurazole and vancomycin (92%), tetracycline (84.6%)	Complete sensitivity to chloramphenicol and amikacin
	Deshmukh et al., 2023 [110]	34	Enrofloxacin (94.11%), tetracycline, lincomycin (85.29% each), cephalexin (70.58%), cefixime (47.06%)	Varied resistance patterns, high sensitivity to gentamicin
	Garhia, 2017 [87]	42	Cefoxitin (97.61%), ciprofloxacin (64.28%), nalidixic acid (33.33%)	Majority resistant to cefoxitin
	Vivekanandhan, 2018 [89]	13	Oxacillin, tetracycline, cefpodoxime (84.61% each), ciprofloxacin (69.23%)	Resistance levels varied among antibiotics
	Begum et al., 2015 [93]	27	Amoxicillin, co-trimoxazole (100%), cephalexin (96.29%)	Majority sensitive to gentamicin
*S. aureus*	Ruban et al., 2018 [112]	80	Ampicillin (100%), tetracycline (87.50%), amoxicillin (77.50%), ciprofloxacin (50%)	Varied resistance patterns, notable susceptibility to gentamicin and vancomycin
*C. perfringens*	Priya et al., 2023 [122]	63	Linezolid (96.83%), clarithromycin (92.06%), erythromycin (88.89%), clindamycin (87.30%), ampicillin (71.43%)	Multidrug resistance prevalent; 100% susceptibility to ofloxacin

## Data Availability

No new data were created or analyzed in this study. Data sharing is not applicable to this article.

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
