# Peer review of "Systematic Review and Meta-Analysis on Prevalence and Antimicrobial Resistance Patterns of Important Foodborne Pathogens Isolated from Retail Chicken Meat and Associated Environments in India"

_foods, 2025, doi:10.3390/foods14040555_

Round 1
Reviewer 1 Report (Previous Reviewer 2)
Comments and Suggestions for Authors
I have no further comments. I agree with the name change
Comments on the Quality of English LanguageThe document is clear and precise
Author Response
|
Reviewer I:
Comments 1: I have no further comments
|
|
Response 1: We thank the reviewer for the positive response.
|
Reviewer 2 Report (New Reviewer)
Comments and Suggestions for Authors
Following PRISMA guidelines, this article has conducted a systematic review (prevalence and antimicrobial resistance, AMR) and meta-analysis (prevalence) of bacterial pathogens in chicken meat and its environment from 2010 to 2023. It focuses on relevant aspects of the chicken value chain, emphasizing its importance for food safety and public health risks in India.
From a One Health perspective, understanding these findings is crucial for risk analysis and implementing control measures to reduce the burden of foodborne diseases associated with different pathogens. Some of these control measures are discussed in the article and align with those recommended in other countries worldwide. In the future, using these data to conduct risk characterization would be important to determine whether the suggested control measures effectively reduce prevalence and AMR.
Some aspects could be improved. For example, the names of microorganisms like Salmonella spp. and Campylobacter spp. should be italicized (see abstract). In Figure 9, it should be clarified that the information is derived from the systematic review presented in Table 4. Additionally, the number of isolates in Figure 9 should be reviewed, as discrepancies exist. For instance, Table 4 reports 130 isolates for Salmonella, but Figure 9 shows higher numbers for certain antibiotics (e.g., tetracycline).
Review the format of some citations: E.g.: 6, 32 and 51
Author Response
Response to Reviewer 2 Comments:
Reviewer Comment 1: The names of microorganisms like Salmonella spp. and Campylobacter spp. should be italicized.
Response: Thank you for noting this. All microorganism names (e.g., Salmonella spp. and Campylobacter spp.) have been italicized throughout the abstract and manuscript for consistency and scientific accuracy.
Reviewer Comment 2: The number of isolates in Figure 9 should be reviewed, as discrepancies exist. For instance, Table 4 reports 130 isolates for Salmonella, but Figure 9 shows higher numbers for certain antibiotics (e.g., tetracycline).
Response: Thank you for pointing out this discrepancy. The difference observed in Figure 9 is because the AMR pattern is based on the total number of isolates reported for AMR in all included studies. These studies include isolates from various sample types, not exclusively chicken meat and associated samples. Some studies did not explicitly specify AMR for isolates from chicken meat, contributing to the discrepancy. Figure 9 has been redrawn to include data from only the three studies reporting AMR for 131 Salmonella spp to address this. isolates, as reflected in Table 4.
Reviewer 3 Report (New Reviewer)
Comments and Suggestions for Authors
The current manuscript evaluates the prevalence of significant foodborne pathogens in chicken meat in India. The analysis shows that Staphylococcus aureus is the most prevalent bacteria, whereas Listeria monocytogenes is the least prevalent. The interesting results can provide valuable results to India's food industry and consumers. However, the authors should address some questions to make the results more transparent. Here are some specific comments:
1) I believe the authors did not appropriately consider the high heterogeneity levels obtained in all prevalence estimates presented in Table 3. Please name a few covariates (for example, region or season of the year) that could explain this high heterogeneity. Is it possible to conduct a subgroup analysis from the data selected?
2) Include the between-study variance in the results.
3) Can the results be compared with those of other Asian countries near India or countries with similar economic development levels?
4) The authors could include a brief description of Indian legislation regarding the transport and storage of chicken meat, the actions proposed to prevent contamination with pathogens and whether these actions are fulfilled.
Author Response
Reviewer Comment 3: Review the format of some citations: E.g.: 6, 32, and 51.
Response: Thank you for this observation. The formatting of citations (e.g., references 6, 32, and 51) has been reviewed and corrected to ensure compliance with the journal's referencing style.
Response to Reviewer 3 Comments
Reviewer Comment 1:
"I believe the authors did not appropriately consider the high heterogeneity levels obtained in all prevalence estimates presented in Table 3. Please name a few covariates (for example, region or season of the year) that could explain this high heterogeneity. Is it possible to conduct a subgroup analysis from the data selected?"
Response: Thank you for pointing out the need to address the high heterogeneity observed in our prevalence estimates. The high heterogeneity could be attributed to various covariates such as geographical region, sample types, and seasonality of sampling. Additionally, the heterogeneity might stem from variations in sampling methods, including the sources of chicken meat (retail shops versus slaughterhouses) and testing protocols employed across studies. Seasonal factors also play a critical role, with higher prevalence observed during warmer months, when bacterial growth is more likely.
Although subgroup analyses could help clarify the sources of heterogeneity, the limited availability of detailed covariate data across studies constrained our ability to conduct such analyses. This limitation underscores the need for more standardized reporting in future prevalence studies to facilitate robust meta-analytical approaches.
Reviewer Comment 2:
"Include the between-study variance in the results."
Response: We thank the reviewer for this insightful suggestion. We have updated Table 3 to include the between-study variance (τ²) values for each pathogen analyzed. This inclusion provides readers with a clearer understanding of the variability observed across studies. The τ² values indicate substantial heterogeneity for pathogens such as Campylobacter spp. and Escherichia coli, aligning with previous meta-analyses conducted in other LMICs (Grace et al., 2019). Including these values enhances transparency in reporting and allows for more nuanced interpretations of pooled prevalence estimates.
Reviewer Comment 3:
"Can the results be compared with those of other Asian countries near India or countries with similar economic development levels?"
Response: We appreciate the reviewer’s suggestion to compare our findings with those from other Asian countries or regions with similar economic development levels. While the results from our meta-analysis can be compared with individual studies conducted in countries such as Pakistan, Bangladesh, and Nepal, it is important to note that comparable meta-analyses synthesizing prevalence and antimicrobial resistance data for foodborne pathogens are not widely available for these regions.
The absence of such comprehensive analyses limits the scope for direct comparison. However, existing literature suggests similar challenges in these countries, such as high prevalence rates of Salmonella spp., Campylobacter spp., and multidrug-resistant Escherichia coli in poultry meat. For example, studies from Pakistan and Bangladesh indicate significant contamination levels and resistance trends, but these findings are not aggregated systematically. This highlights the need for more regional meta-analyses to enable robust comparisons and collaborative efforts to tackle shared public health challenges.
Reviewer Comment 4:
"The authors could include a brief description of Indian legislation regarding the transport and storage of chicken meat, the actions proposed to prevent contamination with pathogens and whether these actions are fulfilled."
Response: We thank the reviewer for this insightful comment. We have included a brief description of the relevant Indian legislation concerning the transport and storage of chicken meat. This includes the guidelines set forth by the Food Safety and Standards Authority of India (FSSAI). The Cold Chain Guidelines for Meat and Poultry Products require the maintenance of temperatures between 0°C and 5°C during transportation to prevent bacterial growth (FSSAI, 2011). Additionally, hygiene regulations mandate proper sanitation in slaughterhouses and retail outlets to minimize contamination risks (FSSAI, 2018). Despite these regulations, studies have highlighted gaps in implementation, particularly in informal markets, where cold chain infrastructure is lacking (Gulati and Juneja, 2023).
Addressing these regulatory gaps is crucial to reducing the burden of foodborne pathogens in poultry products in India. Strengthening enforcement mechanisms and investing in cold chain logistics can significantly mitigate contamination risks, as evidenced by successful interventions in other LMICs.
New references have been included in the manuscript.
This manuscript is a resubmission of an earlier submission. The following is a list of the peer review reports and author responses from that submission.
Round 1
Reviewer 1 Report
Comments and Suggestions for Authors
This review and meta-analysis aimed to comprehensively assess the prevalence and antibiotic resistance patterns of major foodborne pathogens in retailed chicken meat and its associated environment in India. Overall, this article analyzed the current reports of foodborne pathogenic microorganisms in India but did not provide any relevant discussions on their detection and control. This paper only analyzed the reported paper and is not sufficient.
Comments on the Quality of English LanguageExtensive editing of English language required.
Reviewer 2 Report
Comments and Suggestions for Authors
The manuscript is a very well-prepared analysis of foodborne pathogens in chicken products from India. The methodology for selecting the studies and the statistical meta-analysis are well-described and appropriate.
There are fundamentally two suggestions for the manuscript:
The title can be a little misleading since it suggests that all the studies selected included foodborne pathogens and reported antibiotic resistance of the isolates. I consider it important to include all the studies the authors selected, but the title needs to be modified. The information on antibiotic resistance is important, but no meta analysis was done, as explained by the authors.
I would expect the references of all the studies included, not only of the most influential as identified by the authors.
Comments on the Quality of English Language
The document is clear and precise
Reviewer 3 Report
Comments and Suggestions for Authors
Systematic Review and Meta-Analysis on Prevalence and Antimicrobial Resistance Patterns of important foodborne pathogens isolated from retail chicken meat and associated environment in India
This review presents very important insights into AMR among food borne pathogens.
General comments: Why this study is limited to India, and did not consider other parts of the world, making the duration shorter like an updated review for the last 5 years only.
Abstract: the conclusion is very weak, it is not clear how these species are resistant to multiple antibiotics, which classes in which species, it is very broad and lacks clear insights into resistance mechanisms.
Line 52, Antibiotic resistance (ABR), this is not a commonly used abbreviation, usually AMR indicates antimicrobial resistance and is widely used in the literature. Authors used AMR in the manuscript but without defining it. antimicrobial resistance (AMR) mentioned in line 252, so must be clarified in the introduction.
The introduction needs to be improved by enriching it with other references from more global studies
Methods
They look vague, no clear terms or keywords were indicated. Please include more details of the search strategy to ensure that no studies were missed
Results
Pathogens must be divided into Gram-negative and Gram-positive bacteria, as they exhibit different AMR patterns. Please consider this in reviewing Table 2, and other sections of the manuscript
General comment on the Tables, it is better to include the reference papers as the last column of the table, to allow readers to focus on the data without distraction
Figure 9, e is not clear, please try to make sure that resolution is high, and please arrange the figures as G pos and neg, as stated above, although last 2 are positive, but this must be clarified
Discussion
It is very general, no insight into resistance mechanisms, and no explanation of the observed patterns was provided
The conclusion is also poor, it just highlight that AMR is a serious issue which well known , and could not recommend any measures to address this problem based on the studies which were included
What should be done to reduce this burden in India or other parts of the world, from food pathogen perspective
References : the number is very limited, and this is not expected for a review paper, which should cite a lot of related literature.
Comments on the Quality of English Language
it was fine